# Iron-Rich Spherules of Taihu Lake: Origin Hypothesis of Taihu Lake Basin in China

**Shuhao Zuo** [ID] **and Zhidong Xie** *

State Key Laboratory for Mineral Deposits Research, School of Earth Sciences and Engineering, Nanjing University, Nanjing 210046, China; mg1029022@smail.nju.edu.cn
* Correspondence: zhidongx@nju.edu.cn

**Abstract:** In this paper, a detailed mineralogical study on iron-rich spherules in Taihu Lake was carried out, and we present a proposed impact-related origin for these iron-rich spherules. The iron-rich spherical concretions in Taihu Lake occur in a specific silty layer formed around ~7 ka B.P., sandwiched between an upper lacustrine deposit layer and a lower hard loess layer, and they are widely distributed and are the most abundant iron-rich concretions in that specific layer in the vicinity of Taihu Lake. The spherules are typically ~0.5 to 3 mm in diameter with a shape very similar to a spherical shape but not exactly rounded and have various apparent aerodynamic shapes, such as spherical, cone, spindle, ellipsoidal, elongated and pear-shaped morphologies. SEM imaging shows that there is no central core and no concentric layers in the spherules. Iron-rich spherical concretions are similar to accretionary lapilli and have a typical colloidal structure with abundant angular quartz grains and trace fragments of clays wrapped in fine cements that are mainly goethite with minor clays and carbon particles. The typical nodule-forming mechanism in aqueous sediments does not sufficiently explain the morphology and internal features of the iron-rich spherules of Taihu Lake, whereas the aerosol formation mechanism under the airburst impact origin hypothesis of the Taihu Lake basin may be a better explanation of the unique mineralogy of the spherules. Specifically, airburst impact plumes could be the reaction chambers of the aerosol to form the accretionary lapilli with a colloidal texture for the interior, while a dense shell and semi-plastic morphological features can form in the falling processes from higher altitudes in the plume.

**Keywords:** Taihu Lake; iron-rich spherules; goethite; quartz debris; airburst

## 1. Introduction

The origin and evolution of the Taihu Lake basin has attracted the attention of Chinese geologists since the beginning of modern geological studies in China almost 100 years ago. Taihu Lake is the third largest freshwater lake, with a 65 km diameter, in the southeast of China and is located in the center of the triangle of three big cities: Shanghai, Hangzhou and Nanjing, an economic and cultural center of China. Several hypotheses of the Taihu Lake formation were proposed, which include the lagoon hypothesis [1], tectonic hypothesis [2], earthquake hypothesis [3], volcanic hypothesis [4], combination hypothesis [5] and dammed lake hypothesis [6]. However, no hypothesis was seriously studied in detail and in depth. The southwest arc of Taihu Lake leads scholars to doubt that it was formed by a meteorite impact [7–12]. However, the origin of the Taihu Lake basin has still not been confirmed yet.

In the early 1990s, an impact origin was proposed on the basis of fractured quartz and a wavy extinction of quartz grains in the sandstone of the Devonian Wutong Formation, which crops out in the islands of Taihu Lake [7,10]. However, deformation of quartz and the circular structure can have multiple interpretations [13]. The impact origin hypothesis is inconsistent with the very shallow depth (3 m) of Taihu Lake. If the Taihu Lake basin was formed by a meteorite impact, this young Holocene basin should be much deeper

and there should be many pieces of classic evidence of impact such as shocked quartz and high-pressure polymorphs of quartz.

The discovery of unique siderite concretions combined with previous claimed impact evidence revived the impact hypothesis in 2009 [12]. In the dredging project of Shihu Lake in Suzhou, many iron-rich concretions including rod-shaped, spherical and irregular-shaped concretions were found in a specific silty layer [12]. The irregularly shaped ferruginous concretions were regarded as ejecta materials of impact, launched into the air that then fell back into the impact crater and onto the surrounding area [12]. The claim of the confirmation of impact cratering [12] was not confirmed by additional evidence such as a discernable crater rim, a clear central uplift and an Ir anomaly. Additionally, many additional questions remain, which are not addressed in the study of [12], such as what kind of impact mechanism was involved and how these concretions may have formed.

However, the impact hypothesis is still viable because of a substantial amount of interconnected evidence, such as deformation features within quartz grains from the sandstone outcrop of the Taihu Lake area, and the abundance of Fe-rich concretions within one specific marker silt layer of Taihu Lake. All these observed clues, plus the shape and shallow bathymetry of the lake, suggest an airburst impact hypothesis [14], rather than a typical hypervelocity direct contact impact event. Fe-rich concretions may formed in the ejecta plumes resulting from airburst impacts, which could produce a huge, shallow Holocene basin without major crustal disruption under a relatively low but widely distributed shock pressure [13,14].

Iron-rich spherules are the most abundant and widely distributed concretions in the vicinity of Taihu Lake. Engineering projects involving soil extraction from the lake bottom near Xishan Island and other nearby sites also reveal the same specific reduced silty layer in the bottom of Shihu Lake formed around ~7 ka B.P. [15]. The previous literature also mentioned the iron-rich spherules in the Taihu Lake area [6,16–19], and kilometer-long, iron-sand belts were reported at the bottom of the west side of Taihu Lake. The iron-rich spherules were interpreted as the leaching and enrichment product of Fe-Mn nodules from late Pleistocene soil (hard loess layer). The concept of "concretion" mentioned in this paper mainly refers to the colloidal texture of quartz chips embedded in a fine matrix and does not infer the origin of concretion as an aqueous sedimentary product as in the traditional geology context.

This paper mainly focuses on the distribution, occurrence and mineralogy of the iron-rich spherules in the vicinity of Taihu Lake. The goal of this study is to conduct detailed mineralogical and microstructural analysis of Taihu spherules, by comparing them to other Fe-rich nodules formed in lake sediments, to understand and interpret the origin or formation mechanism of iron-rich spherules.

## 2. Geological Background and Experimental Methods

### 2.1. Geological Background and Sampling Location

Taihu Lake is located in the Yangtze River Delta plain and is the third largest freshwater lake in China. The distance from the north rim to the south rim is about 68.5 km, and the width from east to west is about 34 km. The average water depth is 1.89 m, the maximum water depth is close to 3 m, the lake bottom is very flat and the slope is about $0°0'19.66''$, near zero. The southwest shore of Taihu Lake has a semi-circular shape. There are many small islands and two large islands: Dongshan Island and Xishan Island. Dongshan Island has become a peninsula due to the connection to the lake bank in modern times (Figure 1). There are three Holocene units in the Taihu Lake bottom: topsoil, silty layer and hard loess layer (Figure 2). The hard loess layer (unit III upper) extends and connects with the loess layer outside the lake, and above the hard loess, there is a silt and mud layer, which varies from 2 to 4 m [3–5]. Previous studies have shown that the age of the loess layer is from 20 ka B.P. to 11 ka B.P. [20,21]. The iron-rich spherule samples of this study were mainly collected from the silty layer of the lake bottom near Pingtaishan Island, Sanshan Island and Xishan Island and the Shihu Lake bottom, and the samples of the iron-sand

belt were collected at the lake bottom in the middle of West Taihu Lake (Figure 1) near Pingtaishan Island.

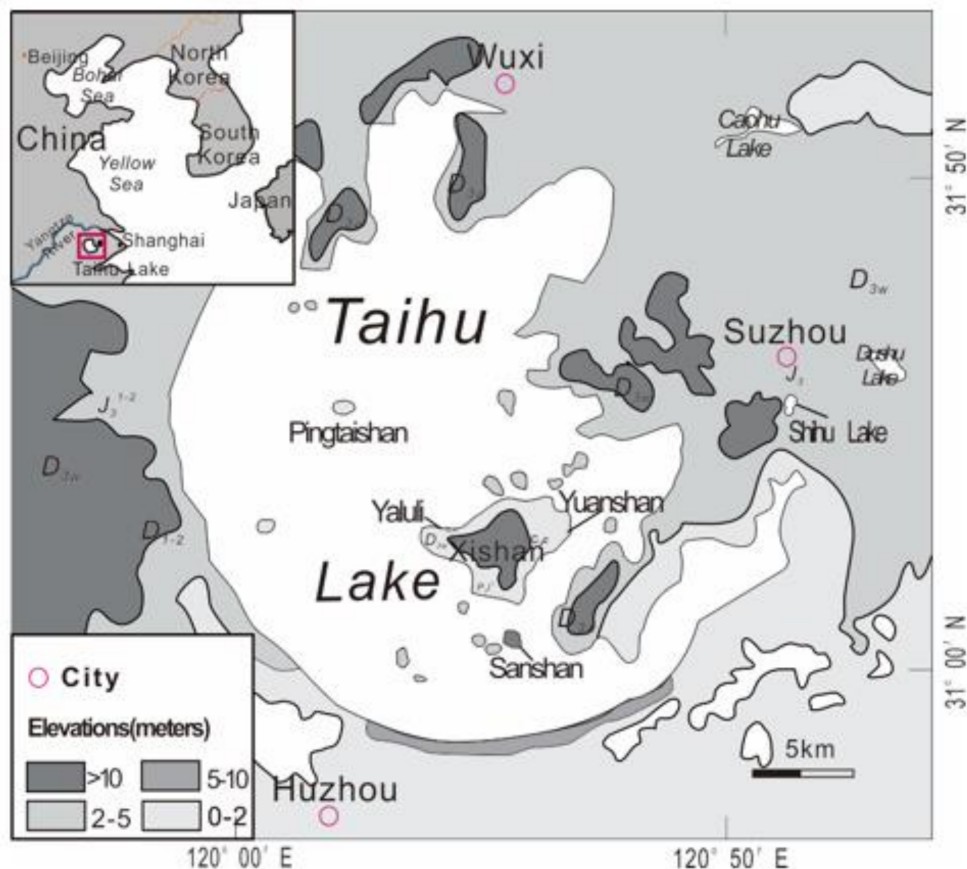

**Figure 1.** Simple sketch of geological map of Taihu Lake area showing sampling sites. White area is lake water, light gray is local plain and dark gray is hilly region. Small map inserted in upper left corner showing Taihu Lake on a regional scale.

The most widely developed bedrocks in the Taihu Lake area are sandstone series of the Devonian Wutong Formation ($D_{3W}$) (Figures 1 and 2). The Wutong Formation mainly consists of a thick, grayish white layer of quartz sandstone with some siltstone, shale and mudstone layers within. There is an intermittent distribution of Carboniferous ($C_3c$) limestone and Permian ($P_2l^1$) limestone, and some Jurassic ($J_3$) volcanic breccia in the islands and surrounding hilly regions (Figure 2). Taihu Lake and its vicinity can be divided into chessboard-like fault blocks by two groups of regional faults in the NW and NE directions in the Indochina period [4]. During the Neogene and Quaternary, both terrestrial and adjacent marine environments existed in the area, and thus the sedimentary thickness of the resulting sedimentary units increases from west to east [6]. The Taihu Lake area is an ice age (Holocene) erosion terrace from 12 k to 20 k years ago, which formed upon a hardened loess layer at a time when the ocean shoreline was about 200 km away (in an easterly direction) from the current ocean shoreline [22,23].

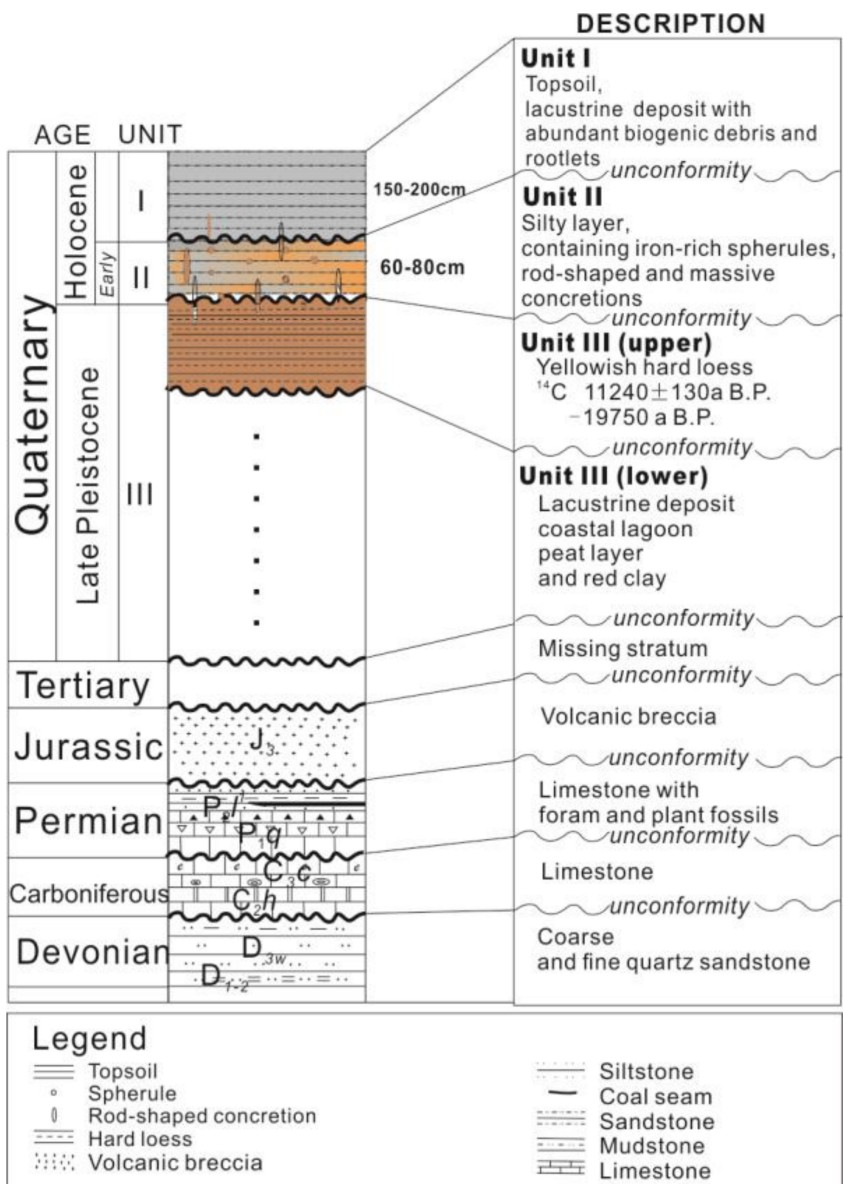

**Figure 2.** Composite columnar section of the vicinity of Taihu Lake area. Unit II is the specific silty layer containing abundant iron-rich concretions, including spherules, rod-shaped concretions and irregular-shaped massive concretions.

*2.2. Experimental Methods*

Various tools were used to describe and investigate the morphology and mineralogy of the samples, which include optical microscopy, scanning electron microscopy (SEM), electron probe microanalysis (EPMA), X-ray powder diffraction (XRD) and transmission electron microscopy (TEM). The observations were mainly carried out in our State Key Laboratory for Mineral Deposits Research, School of Geosciences and Engineering, Nanjing University. Some were conducted in the lab of Arizona State University (ASU), the Guiyang Institute of Geochemistry of Chinese Academy of Sciences (CAS) and the Purple Mountain Observatory of CAS in Nanjing.

Most of the samples reported on in this paper are from Pingtaishan Island, the iron-sand belt under the lake and the bottom of Shihu Lake. There were three ways to collect samples: picking spherules out from exposed piles of the fresh mud layer unearthed up from the lake bottom; extraction from the iron-sand belt under the lake using a specialized iron basket to drag along the iron-sand belt; and drilling the mud core and picking out the

spherules from the mud core. A hand-held, soil sampling drill was used to collect samples in Pingtaishan Island and Shihu Lake. A total of 6 mud samples, each 200 g in weight, from two drilling sites at the depth of 1.5 m, 1.6 m and 1.7 m, were taken out from the core and soaked in water for 24 h. Then, a 50- μm filter screen was used to screen out the small particles and sort out millimeter-sized spherules. Then, the spherules were cleaned by ultrasonic waves and dried at 160 °C. The spherules were immersed in epoxy to produce thin sections or one-sided, polished sections.

The powder of the spherules was prepared by grinding with a ball grinder or an agate grinding chamber for XRD analysis, and some of the powder was used to produce TEM powder samples. In addition, some of the powder was used to extract carbon debris by centrifugal extraction method. XRD analysis was carried out on a Bruker D & Advance Diffractometer. Each powder sample was about 0.5 g. The instrument parameters were: Cu target, K$\alpha$ ray ($\lambda$ = 0.154 nm), Ni monochromator, tube voltage 40 kV, tube current 40 mA, scanning width 2 theta angle from 3° to 70°, speed 0.5°/min. The collected data were analyzed by Jade 6.0 software to provide the mineral components of the samples.

The internal structures of Fe-rich spherules and semi-quantitative mineral analysis were studied by a petrography optical microscope and a scanning electron microscope (SEM), specifically, a JEOL JSM- 6490 (20 kV, 10 nA) with an energy-dispersive spectrometer (EDS) detector. The quantitative chemical analyses of the minerals were carried out by a JEOL JXA8100M electron probe. The EPMA working conditions were as follows: acceleration voltage 15 kV; acceleration current 20 nA; and beam spot diameter 1~2 μm. The data were corrected by the ZAF program. The characteristic peak measurement time was 10 s, and the background measurement time was 5 s. Fe standards used were hematite and ortho-pyroxene, Si standards were quartz and garnet and the Al standard was from garnet.

Three types of TEM samples were produced using focused ion beam (FIB), ultra-thin slicing and powder dripping. In order to achieve the ideal fine particle size of less than a micron for the powder, some samples were added 1 μm-sized $Al_2O_3$ powder to grind together to help make the powder much smaller. We dripped the suspension alcohol liquid containing the fine powder of spherule samples on the carbon film of a TEM carrier to produce a TEM powder sample. The FIB-TEM samples were cut from the shell of the PT09 sample (Figure 6b). The epoxy-wrapped micro-spheres were sliced to produce TEM slices with a thickness of 30–50 nm by using an ultra-thin slicing machine, which then were lied on the carbon film of the TEM carrier. The TEM was an FEI Tecnai-F20 at an acceleration voltage of 200 kv. Select area diffraction pattern (SAED) was used to determine the structure of the minerals. EDS attached to the TEM was used to collect mineral composition data. High-magnification TEM images were obtained to study the detailed microstructures of the colloidal matrix of Fe-rich spherules.

## 3. Results

### 3.1. Occurrence and Distribution of Iron-Rich Spherules

The Fe-rich concretions found on the bottom of Taihu Lake can be grouped into three types: spherules (size from microns to millimeters, and even to centimeters), elongated rods (short to long rods with a diameter of less than 3 cm) and irregular or massive concretions (size from 1 to 15 cm). The irregular shapes include massive, sheet and pear-shaped or teardrop-shaped concretions. All three types occur in one specific silty layer. They were initially found in Shihu Lake near Suzhou city during dredging works [12], and our later work [14,15,24] revealed that there are similar silty marker layers in the vicinity of Taihu Lake, at the lake bottom near Pingtaishan Island, Xishan Island and Sanshan Island and some small islands. The previous literature interpreted them as Fe-Mn nodules formed in aqueous sediments of the hard loess layer, thus determining them as left-over coarse materials after the associated fine silt and mud were washed away [6,20,21]. Fe-rich spherules along with Fe-rich concretions and the marker silty layer are widely distributed in Taihu Lake and the nearby area.

The marker silty layer is sandwiched between yellowish hard loess and modern black silty layers (Figure 3), and the silty layer has a thickness of ~60 to 80 cm. The fresh silty layer is bluish and white, known as green paste mud by local people. If this layer is exposed to the ground for a while, the layer may be partially oxidized rather quickly and turn to reddish-brown (Figure 3a). The middle marker silty layer has more clay, with a more uniform particle size and finer grain size as compared with the upper and lower layers [15]. The millimeter-sized iron-rich lapilli are the signature minerals being seen at almost every location within the marker silty layer, and rod-shaped and irregular concretions were found almost everywhere as well. The age of the specific marker silty layer was determined by C14 dating of charcoal wood and shell fragments in the silty layer as ~7 ka B.P. [15]. In the bottom of a small, adjacent lake (Shihu Lake), one shell layer occurs above the middle silty layer (Figure 3b), which is dated and thus confines the middle silty layer's age to not greater than 6700 B.P. In the Suzhouwan site, a thin brown oxide layer, which is rich in oxidized iron-rich spherules, indicates an unconformity between the upper layer and the middle marker layer [15].

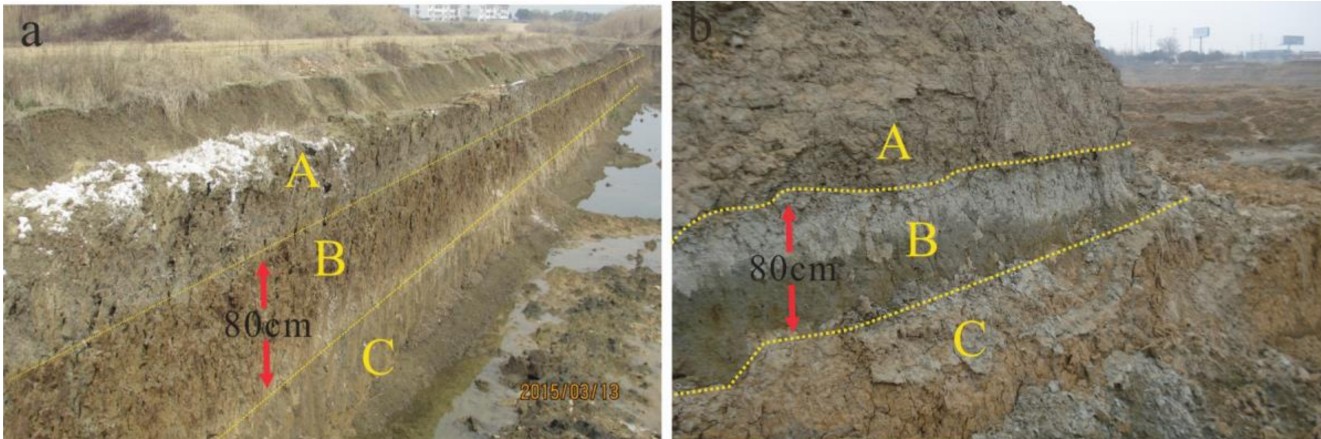

**Figure 3.** The sequence layers of the Taihu Lake bottom (water was temporarily drained away for the excavation, and the area is now covered by water again) near the Yuanshan site of Xishan Island (**a**) and Shihu (**b**). (**a**) The middle marker silty layer (B) is oxidized and shows a reddish-brown color after being exposed to air for many days. (**b**) The fresh lake bottom layers of Shihu near Suzou city. The fresh silty marker layer (B) has a bluish-gray color sandwiched by a lower loess (C) layer and an upper modern mud layer (A).

The fresh sequences can be seen from top to bottom (Figure 3b), if there is no disturbance. The lower layer below the specific silty layer is yellow-brown and contains Ca-concretions, representing a hard loess layer widely distributed in East China. The hard loess layer is a continuous layer in the lake bottom and the surrounding area of Taihu Lake [6,20]. The upper layer above the specific silty layer is from lake deposition with a dark gray-gray color and is rich in plant roots and biochips. However, there are no such sequences in the central location of Taihu Lake. The upper layer and the middle silty layer are almost missing, and only iron-rich sand belts occur above the hard loess layer there [6,17]. The abundant spherules in iron-rich belts were interpreted as the product of washing the lower loess layer by storm flow [6]. Considering similar layers, but more fresh, in different sites, we have a different explanation that iron-rich spherules are from the specific silty layer only, not from the upper or lower layer.

### 3.2. Morphological Characteristics of Iron-Rich Spherules

The sizes of the iron-rich spherules of Taihu Lake vary from microns to millimeters, and rarely to centimeters. This study mainly focuses on millimeter-sized spherules (Figure 4a–d). The shapes of the spherules are approximately that of a sphere, but not a perfect sphere. The smaller the spherules are, the more rounded and more nearly spherical

they are, while the larger spherules typically deviate more from the true spherical shape, as shown in Figure 4b,c. The shapes of the iron-rich spherules vary from spherical, cone-shaped and spindle-shaped to ellipsoidal, teardrop-shaped and rod-shaped. Most of these spherules have what looks to be semi-plastic characteristics. Additionally, their surfaces are covered with small pits. These features are very similar to the shape of the glass tektites and their surfaces as well. All spherules have a black steel to gray, hard outer shell with a thickness of 0.1–0.2 mm. The outer surfaces are both smooth and shiny. The inside of the spherules is relatively soft and comprised of brown, fine granular materials, and a small amount of rock debris. Most of the spherules have many voids under optical microscopy, and some of the spherules contain cavities visible to the naked eye (Figure 4d).

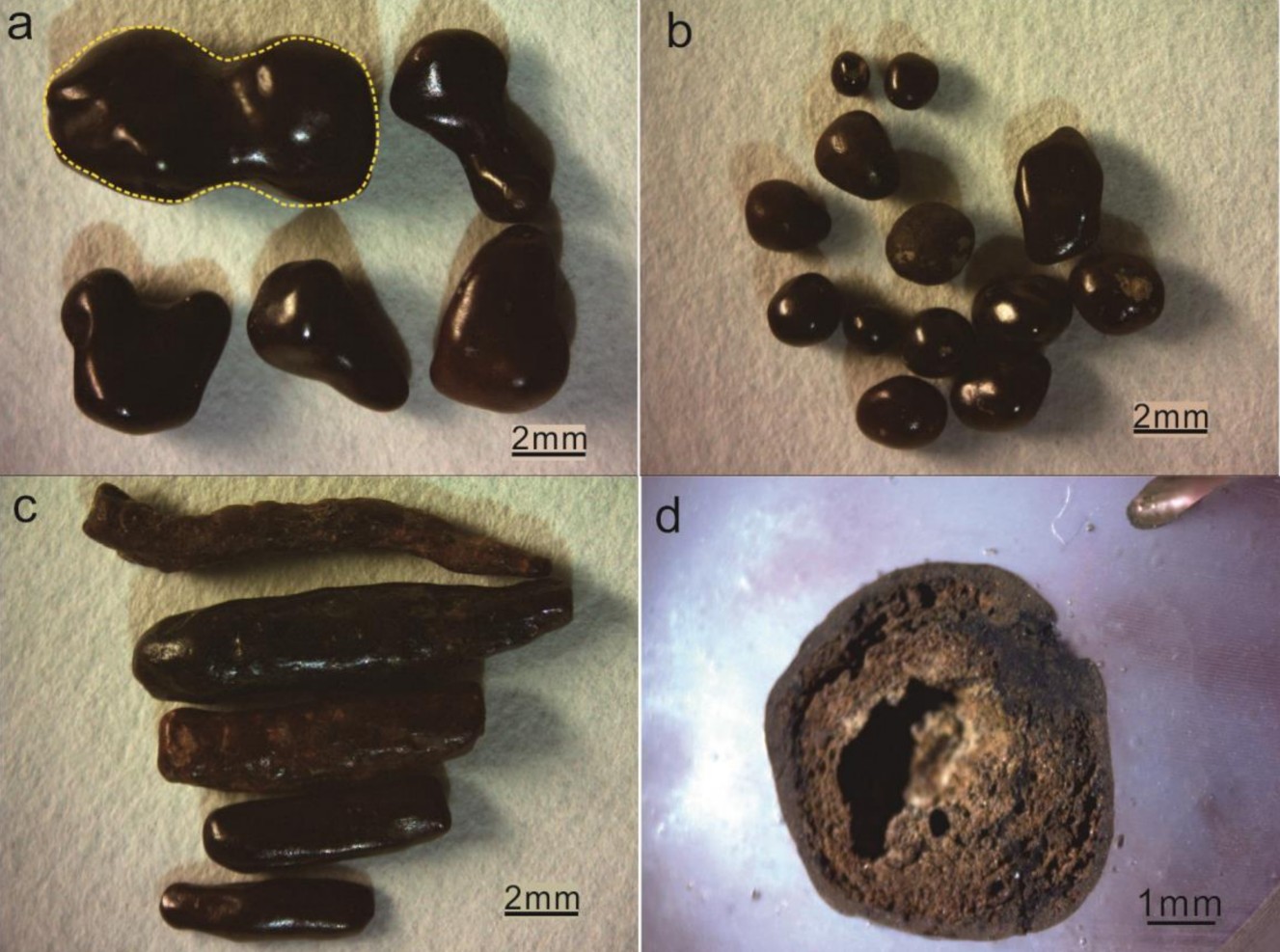

**Figure 4.** Photographs of iron-rich spherules under stereomicroscope. (**a**) The shapes of the iron-rich spherules include ball-shaped, cone-shaped, spindle-shaped, ellipsoidal and teardrop-shaped. (**b**) Millimeter-sized iron-rich spherules with similar and slightly different shapes. (**c**) Small rod-shaped concretions with many pits. (**d**) The cross-section of a spherule showing the structure inside of one spherule with a big cavity and many smaller holes.

*3.3. Microscopic Observation Data*

3.3.1. XRD Analysis of Iron-Rich Spherules

XRD data show that the major minerals of the spherules are quartz, goethite and clay. Additionally, some spherules contain siderite. A total of 16 powder samples were produced for the XRD test, which came from Pingtaishan Island, Yuanshan and Jushan near Xishan Island, Sanshan Island or Shihu Lake. Jade 6.0 software was used for XRD data analysis. The data of five representative samples from the total 16 samples were selected

and are shown in Figure 5. The five samples are labeled as A, B, C, D and E. The data of sample A, B and C were collected from single millimeter-sized grains. Single spherules from samples D and E are too small to have enough powder for XRD analysis; therefore, three small particles from the same site were taken and ground into a powder by using a ball grinder. All XRD data from the 16 samples show that the main minerals are quartz ~50%, goethite ~15%, clay mineral 1%–10% and feldspar 1%–5%. Clay minerals are mainly kaolinite and illite. The data from sample B from Xishan Island have a high kaolinite (K) content (Figure 5). The XRD profile of sample C from Shihu Lake has a siderite peak.

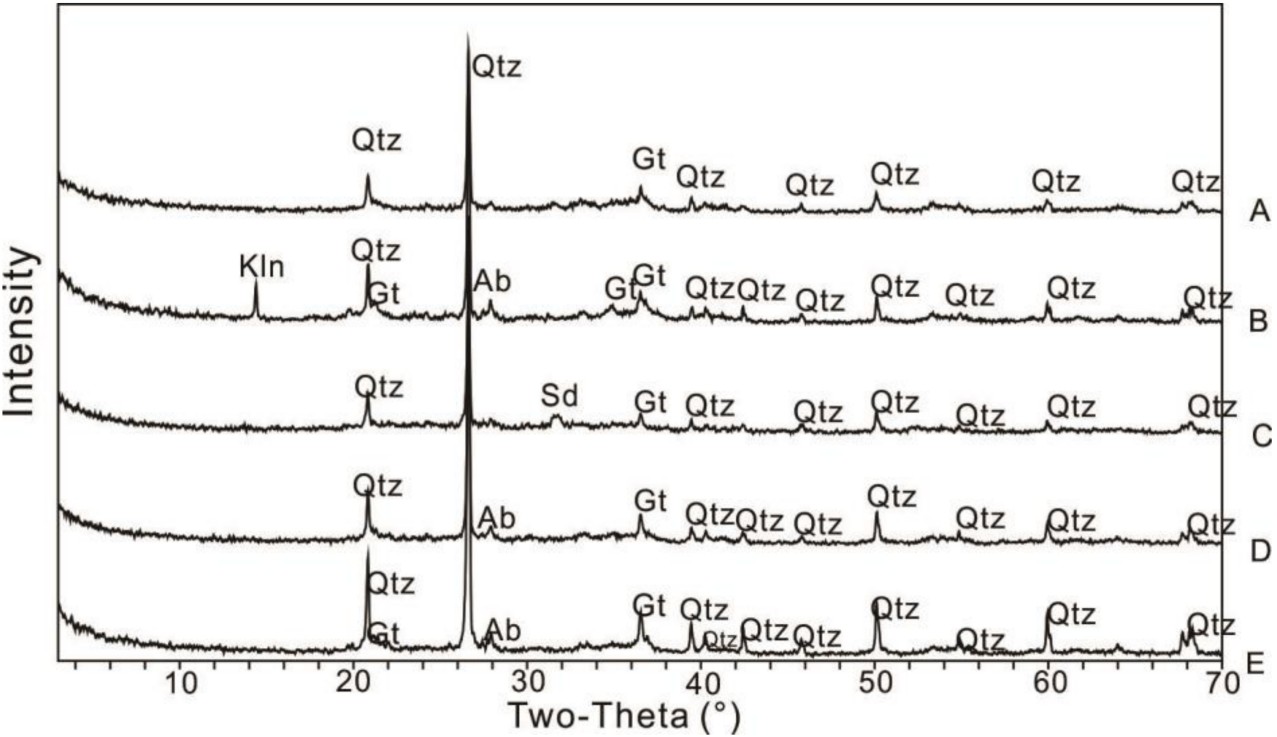

**Figure 5.** XRD profiles from five samples. Samples A and E were collected from Pingtaishan Island, sample B from the lake bottom near Xishan Island and samples C and D from the Shihu lake bottom. Qtz stands for quartz, Kln for kaolinite, Sd for siderite, Gt for goethite and Ab for albite.

The goethite content (10%) calculated by Jade software is much lower than these white-gray parts of the goethite matrix (almost 50% of all) shown in SEM and EPMA images (Figures 6 and 7). This may be due to the varied content of goethite and also because the sizes of the quartz fragments are too small, the amounts of quartz signals are too high and the strong quartz signals reduces the signals detected from much finer goethite and other minerals, which occurs often during XRD analysis [25,26]. In addition, the relevant XRD analysis (not shown here) for the silty layer where these spherules occur shows the surrounding mud is very similar to the data profile of sample B (Figure 5) with more clay minerals (kaolinite K) than the other samples.

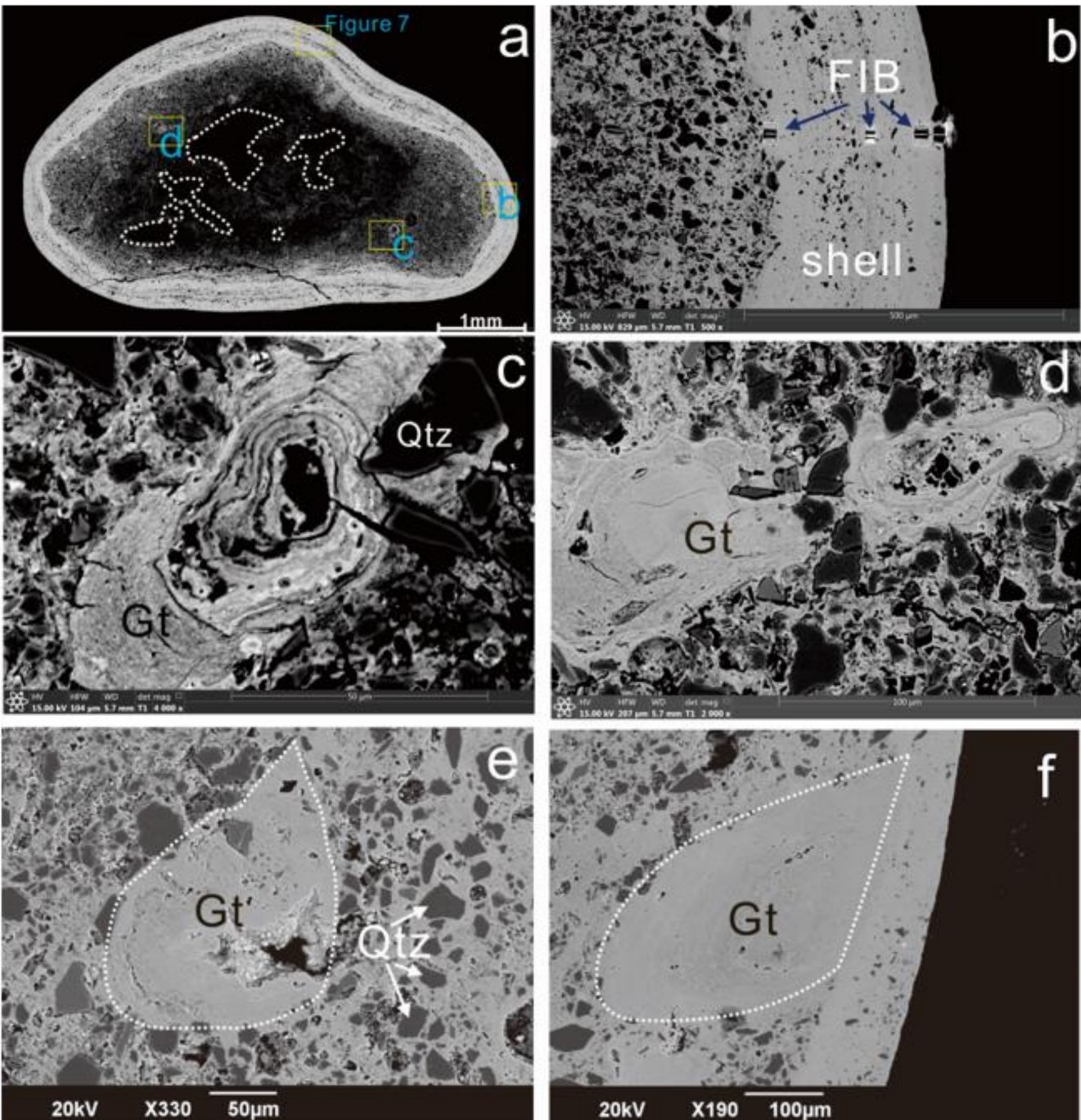

**Figure 6.** BSE images of the iron-rich spherules. (**a**) Low-M BSE mapping image of PT09 shows a dense white shell, many voids (dashed line area) and sites of Figures 6b–d and 7. (**b**) BSE image in b site of Figure 6a showing the interior has less cement with a dark color and more quartz debris (dark gray), and four FIB sites. (**c**) Colloidal grain with concentric layers embedded in PT09 which is located in c site of Figure 6a showing squeezed and deformed features extruded by a quartz grain. (**d**) BSE image of the colloidal grains embedded in PT09 which are located in site d in Figure 6a showing obvious semi-plastic morphology. (**e**) Tear droplet-shaped cement grains with less quartz debris embedded in the PT14 sample. (**f**) Teardrop-shaped cement grain embedded in the PT14 sample. Qtz = quartz; Gt = goethite.

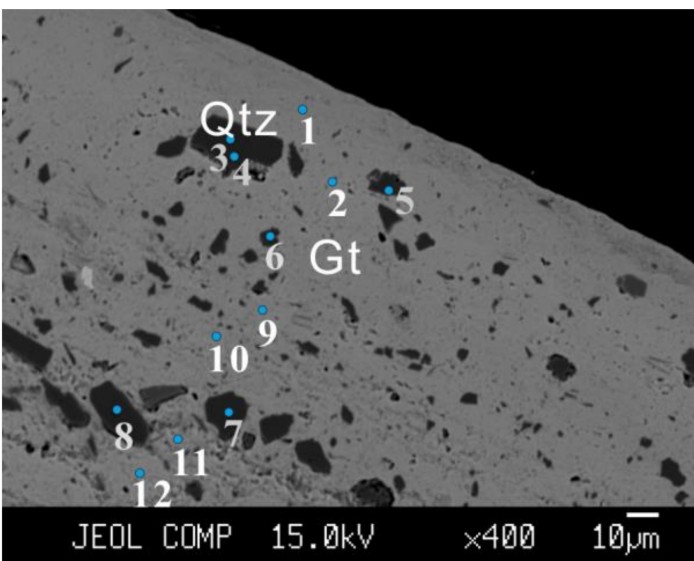

**Figure 7.** BSE image showing the electron spots of EPMA in PT09 sample nearly along one line from outside rim to inside core. Most of the dark gray grains are quartz grains, and the bright gray part is the fine matrix of goethite.

### 3.3.2. Micro-Textures of Iron-Rich Spherules

The results of the SEM and optical microscopy are consistent with the XRD data. The iron-rich spherules are similar to lapilli, mainly containing angular quartz particles as inclusions and an iron-rich fine matrix as a cement. Here, we mainly show the data from PT09 samples from Pingtaishan Island to describe the micro-textures of the iron-rich spherules, which can represent typical spherules. PT09 has an ellipsoid shape, 3 mm × 5 mm in size, with a dense white (high-atomic weight) shell with a thickness of ~0.2 mm, as shown in the SEM-BSE image (Figure 6). The interiors of the spherules have a darker color with a much looser texture and a higher quartz content, and many voids or holes (Figure 6a), which is shown in the hand specimen too (Figure 4d).

SEM images show that both the coating and interior of spherules have a typical colloidal texture with debris wrapped and cemented by fine matrix crystals. No obvious ring band structures and central nuclei were found in spherules. The particles of quartz are very angular. TEM data and XRD data confirm that the fine matrix crystals are microcrystalline goethite. The shell has more denser iron-rich cryptic cements and fewer amounts of quartz chips. The low- and high-magnification BSE images show more detailed textures of the coating shell of the spherules with a thickness of ~200 µm (Figure 6a,b). There are many fine layers with a concentric layer, and the long axis of quartz particles is nearly parallel to the fine layers (Figure 6a).

The internal structure of the lapilli has a typical colloidal texture too. The cement matrix consists of cryptic goethite microcrystals. There is a small amount of clay and felsic minerals. The angular quartz grains with significant angularity and good sorting are randomly dispersed in the spherules' interiors (Figure 6b,e,f). The sizes of quartz particles vary, which can be as low as 2–3 µm but are 10–20 µm in general, and some can be larger at about 50 µm. Some colloidal particles with fine concentric rings are found embedded in the matrix of the spherules (Figure 6c,d). These embedded particles have hollow textures as well, and they have a lower quartz content; further, they show squeezing features along the boundaries extruded by quartz particles (Figure 6c,d). In addition, there are many smaller tear-shaped colloidal spherules embedded inside of the spherules' interiors (Figure 6e,f).

### 3.3.3. EPMA Data

EPMA data are consistent with the SEM and XRD data in general, showing that the major minerals in spherules of Taihu Lake are fine goethite cementing angular quartz

debris with minor minerals of clay and feldspar. Quantitative analyses of EPMA show more details about some minor elements other than major elements (Table 1). EPMA data of 12 points from the outside rim to the inside of the PT09 spherule were collected and analyzed (Figure 7, Table 1). The data of points 1, 2, 9, 10, 11 and 12 are from Fe-rich matrix (bright color in the BSE image) showing major elements of cements are Fe and O with a certain amount of Si and Al. The total amount ranges from 83% to 91%, which can be explained by the hydrogen content in goethite. The content of FeO in the fine matrix increases from 50 for the inside points (points 9, 10, 11 and 12) to 72 wt% for the outside points (points 1 and 2). The content of $SiO_2$ and $Al_2O_3$ is decreased from 21 and 16 for the inside to 4 and 8 wt% for the outside points, but this is still quite a lot in a goethite standard. We cannot rule out that the Si and Al may come from other clay or feldspar minerals mixed in the matrix due to the resolution limitation of the electron microprobe. The TEM technique is needed to address the question of whether these Si and Al are from goethite or not.

**Table 1.** EPMA analysis of PT09 (weight %).

| Element | 1 (Gt) | 2 (Gt) | 3 (Qtz) | 4 (Qtz) | 5 (Qtz) | 6 (Qtz) | 7 (Qtz) | 8 (Qtz) | 9 (Gt) | 10 (Gt) | 11 (Gt) | 12 (Gt) |
|---|---|---|---|---|---|---|---|---|---|---|---|---|
| $K_2O$ | 0.05 | 0.06 | 0.01 | 0.01 | 0.00 | 0.01 | 0.00 | 0.00 | 1.05 | 0.98 | 0.99 | 1.99 |
| MgO | 0.03 | 0.06 | 0.00 | 0.00 | 0.00 | 0.00 | 0.00 | 0.00 | 0.37 | 0.38 | 0.00 | 0.00 |
| FeO | 71.26 | 72.43 | 1.29 | 1.47 | 1.60 | 1.33 | 1.33 | 1.30 | 60.86 | 62.66 | 58.36 | 50.51 |
| CaO | 0.06 | 0.03 | 0.00 | 0.00 | 0.00 | 0.00 | 0.00 | 0.00 | 0.10 | 0.11 | 0.00 | 0.00 |
| $Al_2O_3$ | 7.60 | 7.49 | 0.00 | 0.00 | 0.03 | 0.08 | 0.12 | 0.02 | 11.73 | 12.18 | 13.51 | 16.17 |
| MnO | 0.14 | 0.13 | 0.00 | 0.00 | 0.01 | 0.00 | 0.01 | 0.01 | 0.17 | 0.14 | 0.09 | 0.10 |
| $TiO_2$ | 0.08 | 0.18 | 0.00 | 0.00 | 0.00 | 0.00 | 0.00 | 0.07 | 0.25 | 0.23 | 0.27 | 0.36 |
| $SiO_2$ | 4.15 | 2.56 | 90.52 | 95.66 | 90.98 | 90.94 | 90.73 | 92.25 | 13.00 | 14.00 | 16.28 | 20.66 |
| $P_2O_5$ | 0.84 | 0.83 | 0.00 | 0.04 | 0.00 | 0.00 | 0.01 | 0.01 | 0.49 | 0.52 | 0.00 | 0.00 |
| $ZrO_2$ | | | | | | | | | | | 0.03 | 0.04 |
| Total (wt.) | 84.19 | 83.76 | 91.82 | 97.17 | 92.62 | 92.35 | 92.21 | 93.66 | 88.02 | 91.19 | 89.54 | 89.83 |

(Notes: Gt = goethite; Qtz = quartz. Standards used for Fe are hematite and ortho-pyroxene; Si: quartz and garnet; Al: garnet.)

The totals of EPMA data from clastic quartzes are less than 100% too, from 92 to 97 wt% for points 3, 4, 5, 6, 7 and 8 (Figure 7 and Table 1). The standard sample of quartz was used to double-check the quality of the data, and the total is 98.27–98.99%, indicating the data are effective. In addition, after the 20-s microprobe collection, the electron spots on quartz grains show a millimeter-sized blur area due to beam damage. Under the same conditions, we tested the other quartz sample and standard quartz sample, and there was no beam damage. The reasons why the quartz does not have totals close to 100% remain unknown.

### 3.3.4. TEM Data and C Bits

TEM was used to study the microstructures and micro-minerals of the cement matrix. Three TEM samples were produced including powder TEM samples, FIB-TEM samples and ultra-thin slice TEM samples. TEM data show that the matrix cements of spherules are 10–20 nm-sized goethite microcrystalline aggregations (Figure 8a,c,d). TEM EDS data collected by using a nanometer-sized electron beam show that the iron-rich matrix mainly consists of goethite (Figure 8a,c) with mole percentages of Fe of ~33%, Si of ~6%, Al of ~4% and Mn of ~2%, and a small amount of K at ~0.2% and Ca at ~0.4%. TEM EDS data were collected from a very small area excited by the transmission electron beam on a nanometer-sized diameter area; therefore, we can exclude quartz particles or other visible clay mineral particles in the collection area. TEM-EDS data show that goethite minerals do contain a certain amount of Si and Al in the goethite crystal structure, which is related to the pH and temperature condition of the goethite formation [27].

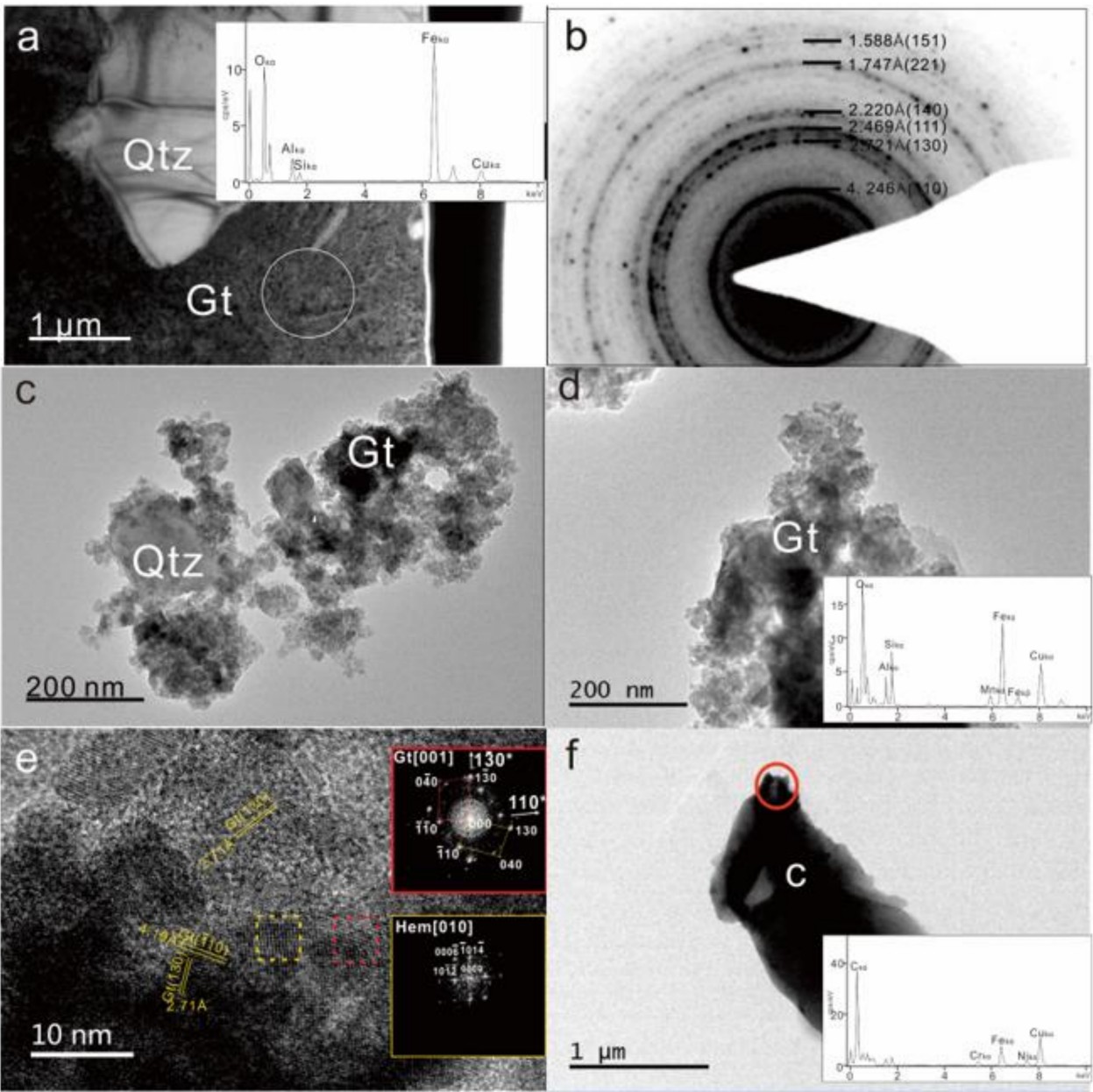

**Figure 8.** TEM images of iron-rich spherules. (**a**) Bright-field (BF) FIB-TEM image of PT09 sample showing quartz grain and goethite microcrystalline aggregations. TEM-EDS profile inserted in lower right corner is collected by using electron beam of spot size of 500 nm shown as a circle at bottom right of the image. (**b**) SAED ring pattern of PT09 TEM sample collected from goethite aggregation showing the D-spacing values for each ring which match goethite; the numbers on the left are the measured D-spacing values, and numbers in brackets are the planes indexed. (**c**) BF TEM images of powder TEM sample of PT02-02 showing quartz grain and goethite aggregations. (**d**) BF TEM images of powder TEM sample of PT02-04. TEM EDS profile is inserted in the lower right corner, which was collected from the center upper edge area less than 100 nm in diameter. (**e**) High-resolution TEM images of center upper edge region in Figure 8d showing lattice fringes of goethite and hematite crystals. Inserted image in upper right corner is a reduced forward Fourier transformation (FFT) image obtained from red dotted square area showing goethite [001] zone axis. Inserted lower right image is FFT image from yellow dotted square area showing hematite [010] zone axis. (**f**) BF TEM images of powder TEM sample of PT02 showing a carbon chip containing Fe-Ni-Cr alloy in upper right corner of the chip. TEM-EDS profile is inserted in the lower right corner showing certain amount of Fe-Ni-Cr in a C background. Gt = goethite, Qtz = quartz, Hem = hematite.

The SAED of the PT09 sample shows a diffraction ring pattern (Figure 8b). D-spacing values and brightness intensities of each ring are consistent with the goethite XRD data.

Goethite in the matrix is confirmed based on the data of diffraction rings. The D-spacing values are close to these indexes using iron-rich chlorite, but the variation in the brightness intensity of each ring does not match with the iron-rich chlorite. By using unit cell software regression calculation, the cell parameters of goethite can be obtained as a: 3.040 Å; b: 10.089 Å; c: 4.657 Å cell; and Vol: 142.85 Å$^3$, with a matching degree of 95% to the standard goethite with a: 3.022 Å; b: 9.955 Å; c: 4.616 Å, and cell Vol: 138.91 Å$^3$. HRTEM images show that most parts are goethite aggregations with minor hematite microcrystalline too (Figure 8e).

Certain amounts of C bits were found in the TEM results (Figure 8f) for the powder TEM sample, which is consistent with the XRD data showing a certain peak of amorphous carbon. A large amount of carbon bits was collected from the powders of iron-rich spherules by using the centrifugal separation method. Carbon chips often contain various minerals with a size ranging from a few micrometers to millimeters. One small carbon chip with a size of 2 μm in length and 1 μm in width was found containing Fe-Ni-Cr alloy (Figure 8f). One large carbon chip over 30 μm in length was found sitting on the TEM sample holder grid. The abundant existence of carbon bits in TEM samples is consistent with the high content of total organic carbon (TOC) in the marker silty layer [15], which may infer a strong reductive nature of the environment when the spherules formed.

## 4. Discussion

### 4.1. Iron-Rich Spherules Occurring in A Marker Silty Layer

Our study shows that there is one marker silty layer formed around ~7 ka B.P. [15,28] which has very different sedimentary characteristics compared with the upper and lower layers. The marker silty layer has uniform silty sizes, higher total organic carbon (TOC) contents, higher magnetic susceptibilities and different XRF element content profiles. The lower layer represents the hard loess layer with an aeolian origin after the last ice age at ~11.7 ka B.P. when the Taihu Lake basin had not formed yet. The upper layer represents the modern Taihu Lake deposit layer after ~5 ka B.P. [15]. The marker silty layers widely occur in the Taihu Lake area, containing abundant iron-rich concretions. These iron-rich concretions can be divided into three categories based on morphology: spheroid concretions (here referred to as spherules), elongated concretions and irregular-shaped or massive concretions. All concretions have a typical colloidal structure with abundant angular quartz grains and minor fragments of clay and feldspar embedded in the fine matrix of siderite crystalline aggregations in elongated and massive concretions or goethite crystalline aggregations in spherules. Three types of iron-rich concretions occur together and have intergrowth relationship. Sometimes, they are interconnected with each other, such as through inter-wrapping, inter-including, inter-winding or inter-crossing, which suggests they formed at nearly the same time in one event.

The occurrence of spherules in the marker silty layer and the unique upright occurrence of rod-shaped concretions indicate that they are not erosional lag deposits from the underlying hard loess layer, which was suggested by other scholars [6]. Many previous studies also found there are abundant spherules occurring at the bottom of the Taihu Lake basin [6,18,19]. The iron-sand belts near Pingtaishan Island were reported [6] and are thought to be accumulating belts of iron-manganese nodules which were washed out from the underlying hard loess layer, and later accumulating as iron-sand belts by storm flows in the center of the lake.

Our studies do not agree with these previous explanations. We think they are accumulating products of seasonal storm flow, but from the iron-rich marker silty layer, not from the underlying hard loess layer. Our previous works discovered that the original layers in Shihu Lake and East Taihu Lake are without or with little later perturbation after deposition [14,15]. There are primitive marker silty layers with rod-shaped concretions vertically occurring in the silty layer. No similar iron-rich concretions, except some calcareous ginger stone nodules, are found in the underlying hard loess layer and the upper modern mud layer. The vertical occurrences indicate that the iron-rich concretions formed later than the

marker silty layer or at the same time. The spherules and the rod-shaped concretions bond with each other and occur together, even with some elongated concretions consisting of many small spherules, and often seen spherules attached to the surface of the rod-shaped concretions [12].

*4.2. Iron-Rich Spherules Are Not Iron-Manganese Nodules*

The spherules of Taihu Lake are more like accretionary lapilli, which are obviously different from typical iron-manganese nodules. Most of the iron-manganese nodules have a nucleus surrounded by zoned Fe and Mn oxide rings from inside to outside [29]. Our observation shows that no such center core exists in the spherules of Taihu Lake, and no Fe-Mn oxide ring zones occur too, except in the outermost shell coating layer (Figure 6a,b). The growth of Fe-Mn nodules is typically formed in the pore space of sediments under the water table within the reductive soil or mud [30]. When in the reduction condition, Fe and Mn with higher electrovalence in oxides were reduced to $Fe^{2+}$ and $Mn^{2+}$, and $Fe^{2+}$ was rapidly oxidized and deposited on the surface of $MnO_2$ under the catalysis of $MnO_2$; when the soil became dry and was under the oxidation condition, $Fe^{2+}$ and $Mn^{2+}$ were oxidized and deposited on the surface of Fe-Mn oxides. The alternations of dry and wet result in the repeated redox and cause the formation of Fe-Mn nodules with rings [29].

The iron-rich spherules of Taihu Lake are not the traditional limonite nodules with a central core and radial textures, which typically form in the pore spaces of aqueous sediment environments [26]. Most of our spherules have a typical colloidal structure. Except for the coating layer, the whole spherule has abundant angular quartz grains and some rock debris, and even many internal hollows cemented by a fine goethite matrix (Figure 6). There is no zoning of Fe and Mn, no obvious central nucleus and no radial structure in the spherules. The cement matrices are goethite microcrystals with a size of several tens of nanometers which form small aggregates, as shown in the HRTEM image (Figure 8a–d). Some heavily oxidized spherules show some concentric ring textures, which were interpreted as the products of biological deposition, formed by the gradual centralization of iron by bacteria in the silty layer [31]. They are obviously heavily oxidized by later oxidation on the shore for a while after salvage or in dry conditions. In some instances, we see that some soft spherules or rod-shaped concretions have a fresh plant in them, meaning some contamination of the current environment. Therefore, we think the samples in [31] were heavily oxidized and contaminated and were not as fresh as the samples we collected. Our fresh spherule samples collected from the buried marker silty layer actually have some reduced iron element, $Fe^{2+}$ or even metal Fe, as shown in Figure 8f, which indicates a non-traditional oxidation environment for forming goethite, rather than one reductive environment.

*4.3. Iron-Rich Spherules Are Aerosol Products Rather Than Hydrosol Products*

Iron-rich spherules from the Taihu Lake area have colloidal textures with a matrix of nanometer-sized goethite crystals cementing abundant quartz particles and a small amount of kaolinite, feldspar and rock debris, which suggests that the spherules of Taihu Lake can be either a hydrosol product or an aerosol product. Solid sol products are another possibility but are not realistic in the background of Taihu Lake and not considered further here.

The shapes of spherules are not perfectly spherical and include many similar shapes to a spherical shape, such as cone, spindle, ellipsoidal, rod/stick-like and teardrop (Figure 4). The actual sizes of spherules range from microns to millimeters, and to centimeters for some cases, but, here, we focus on millimeter-sized spherules. Most spherules are detached, while some of them are aggregates of smaller spherules. Some massive concretions and some rod-shaped concretions are aggregates of many spherules, being presented in other papers [12,14]. The surface topography of spherules varies from smooth curved surfaces (Figure 4) to uneven surfaces with many pits (Figure 4c). Many spherules have hollows inside (Figures 4d and 6a). There are many small nuggets inside of the spherules with semi-plastic features (Figure 6c–f). Some nuggets have a concentric ring texture (Figure 6c,d),

and some do not. It is difficult to explain why they form these topological surfaces in a hydrosol condition, typically requiring a still aqueous environment.

EPMA data and EDS data from TEM and SEM all show that the goethite matrix is rich in Si and Al. The goethites in the outside rim are dense and have higher Si and Al contents than those in the inside (Figure 7 and Table 1). Detailed TEM EDS data collected from a nanometer-sized area show that Si and Al are not from mixed quartz or clay but come from the goethite itself. That means goethite in the matrix contains some Si and Al in its crystal structure, which may offer us some information about the formation condition of goethite. The amount of Si and Al in goethite changes along the temperature at a certain pH [27]. At the low-T or normal hydrosol condition, it is difficult for Si and Al to enter the goethite lattice, but at a higher T, such as in the aerosol condition, more Si and Al are found in goethite crystals.

The coating layer of the PT09 spherule has a thickness of ~200 μm, with a denser goethite matrix, and small quartz grains follow the curvature of the spherule coating and form vague thin layers, which suggests accretion, rather than concretion, as in sedimentary digenesis [29]. In a simple concretion, the small quartz grains might follow the original layering of the sediment and thus go across the spherule grain, not being part of its structure. The inside core of the spherules is more porous with many deformed voids and semi-plastic nuggets, and angular quartz particles randomly distributed in the matrix of goethite, which are not typical characteristics of hydrosol colloids either.

*4.4. Origin of The Iron-Rich Spherules and Airburst Plume*

The origin of iron-rich spherules in the marker silty layer of ~7000 B.P. in Taihu Lake provides a clue concerning the formation of the basin of Taihu Lake and may hold a record of the geological event at that time. The origins of spherule concretion can be sedimentation [29,32] formed in hydrosols, which is not favored, as discussed in the last section, volcanism [33–35] or an impact-related event [36,37]. Volcanism and impact plume can provide an aerosol environment [38–41]. The morphology of the spherules of Taihu Lake is similar to the volcanic lapilli with an aerodynamic shape. Volcanic lapilli generally contain volcanic components, which are lacking in the spherules of Taihu Lake. In addition, no volcanic eruption during the Holocene has been recorded in the Taihu Lake area, suggesting they are not volcanic lapilli.

The impact plume can form a gas column full of debris of the target and projectile materials in impact crater processing, which later condense from the plume and fall down as a fallout layer onto the ground [35–37,42,43]. There are certain amounts of spherules with $CaCO_3$ spherules in the K/T boundary related to the Chicxulub crater [34,36,37,42]. The $CaCO_3$ spherules in the K/T boundary were interpreted as the fallout of an impact plume, which suggests that $CaCO_3$ was synthesized during the impact plume [37], suggesting that the $FeCO_3$ siderite in the rod-shaped concretions of Taihu Lake may have formed in an impact plume too [14]. Therefore, iron-rich spherules may have formed in a plume too.

Traditional contact impact cratering is not applicable in the formation of the Taihu Lake basin, due to the lack of large-scale molten materials, contradicting the basic characteristics of Taihu Lake, being large, flat, shallow and young [13,14]. We hypothesize that multiple meteorite or comet airburst impacts may have disturbed the shallow layer of the Taihu Lake area and formed impact plumes with aerosol environments which contained lots of fine debris including fine soil dust which came from the topsoil layer of the hard loess rich in $Fe^{2+}$ and $Fe^{3+}$, and fine quartz grains coming from the bedrock of sandstone or the topsoil layer. The impact plume could be the reaction chambers of the aerosol to synthesize the goethite and form spherules similar to the accretionary lapilli with a colloidal texture for the interior, while a dense shell and semi-plastic morphological features can form in the falling processes from higher altitudes in the plume. The fallout of impact plumes may include a marker silty layer and iron-rich concretions [14]. The unique shape and internal characteristics of the spherules of Taihu Lake may have formed during the initial formation of spherules and the later falling processes from the plumes.

The iron-rich spherules of Taihu Lake are not volcanic lapilli, nor are they traditional Fe-Mn nodules, or aqueous deposition and hydrosol products. On the other hand, they are similar to accretionary lapilli, more likely formed in an aerosol environment and can be considered fallouts from the impact plume, which might lead to the hypothesis of the origin of the Taihu Lake basin being an airburst impact. The airburst hypothesis of Taihu Lake needs to be further studied.

**Author Contributions:** Conceptualization, Z.X.; Data curation, S.Z.; Funding acquisition, Z.X.; Investigation, S.Z. and Z.X.; Writing-original draft preparation, S.Z. and Z.X.; Writing-review and editing, S.Z. and Z.X.; Project administration, Z.X. All authors have read and agreed to the published version of the manuscript.

**Funding:** This research was funded by NSFC (40972031) and NSFC (41272057).

**Acknowledgments:** Thanks to Laijing and Jiachao for providing samples for this study. Many thanks to Tom Sharp at ASU for critical comments and helpful discussions. Thanks to the reviewers for helping and constructive comments. Thanks to the editor of *Minerals* for inviting us to submit this paper.

**Conflicts of Interest:** The authors declare no conflict of interest.

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
