# Peer review of "Iron-Rich Spherules of Taihu Lake: Origin Hypothesis of Taihu Lake Basin in China"

_minerals, doi:10.3390/min11060632_

Round 1

Reviewer 1 Report

This is an excellent manuscript, rich in data that are subject to careful analyses. It is very appropriate to the journal, and the topic and will be of wide interest to readers. I am pleased to recommend it for acceptance pending very minor modifications. Most of my review comments consist of suggestions for improving the wording of some sentences. I have a few more general concerns.

The objects that are interpreted as products of impact-related airburst plumes have iron-rich compositions, but the bedrock of the region is primarily sandstone and limestone. It would seem likely that material incorporated in the impact plumes would come from these low-iron sediments, not from iron-rich strata that constitute only a minor constituent of the stratigraphy. This issue deserves discussion.

The mineralogy of the iron-bearing granules puzzles me. The XRD patterns are used to identify the presence of goethite as the primary iron mineral. However, XRD patterns do not reveal the presence of amorphous minerals. In sedimentary environments, hydrous iron oxides (e,g, limonite family minerals) are common, but not detectable by X-ray diffraction. In this report, the XRD calculations showed a goethite content of 10% (line 251), but the SEM/EDS and EMP results show much higher Fe values. This suggests to me that the XRD patterns are failing to show the presence of non-crystalline iron minerals. This is another topic that requires some discussion.

My other comments all involve very minor issues:

Title:  should refer to Tiahu Lake, China so that international readers will clearly know the location.

Lines 28, 52: the reference to siderite concretions is confusing, since the masses described in this report contain very little siderite. I suggest a better description would be “ferruginous concretions”.

Line 67: “most abundant” should be “the most abundant”.

Line 113: “Drilling” should be “drilling”.

Line 114: “agate” is a mineral name. I suspect the correct term here is “agate mortar” or “agate grinding chamber”.

Line 166: “diffraction patter” is a typing error for “diffraction pattern”.

Line 174:  The size range is needed for the irregular or massive concretions, to match the description of other concretion forms.

Lines 138, 190, 233: “Bean sized” is an uncertain description, because various bean species have a very wide size range.  In line 448, the description is “Bean-mm-sized”. I suggest eliminating “bean” as a size category, and instead make reference to actual metric dimensions.

Line 405 refers to “ginger-shaped” concretions, a description that will lack meaning for many readers.

Line 254: the wording needs to be re-considered.  As presently phrased, the statement is that the signals from quartz are too strong to reduce the signals from goethite. This is the reverse of the intended meaning: the strong quartz signals reduce the signals detected from goethite.

Table1:  The values need to be labelled as “weight %”, not “%”

Line 377: “TOC” is described as an abbreviation for “total of carbon”. The correct definition is “total organic carbon”.

Line 451: “kind of topological surfaces…..” would be clearer with something like “The surface topography of spherules varies from smooth curved surfaces (Fig. 4) to uneven surfaces with many pits (Fig. 4c).”

Line 471: “more loose” is a questionable descriptions. Perhaps you mean “more porous”?

Line 486: “certain amount spherules” should be “certain amount of spherules”.

Line 494: “we hypotheses” should be “we hypothesize”.

Figure 4: The yellow scale bars for images a,b,c are difficult to read. Changing to black color would help, in addition to increasing the font size.

These are all very minor issues. There are some other examples, but I suspect that the copy editing process will be a place to address improvements. My main point in this review is to express my belief that this is an excellent manuscript that deserves to be accepted for publication.

Author Response

Open Review 1st:

  1. The objects that are interpreted as products of impact-related airburst plumes have iron-rich compositions, but the bedrock of the region is primarily sandstone and limestone. It would seem likely that material incorporated in the impact plumes would come from these low-iron sediments, not from iron-rich strata that constitute only a minor constituent of the stratigraphy. This issue deserves discussion.

Response: This is a very good comment. The bedrock is primarily sandstone and limestone. The topsoil in this region is primarily hard loess layer rich in Fe2+ which might be the main source for the iron of the spherules. We added some text in the discussion part to emphasis this point in page 16, line 493-496. The idea is that the plume included topsoil dust and quartz fragments which might come from hard loess rich in Fe and bedrock which can provide quartz and CO2. . 

  1. The mineralogy of the iron-bearing granules puzzles me. The XRD patterns are used to identify the presence of goethite as the primary iron mineral. However, XRD patterns do not reveal the presence of amorphous minerals. In sedimentary environments, hydrous iron oxides (e,g, limonite family minerals) are common, but not detectable by X-ray diffraction. In this report, the XRD calculations showed a goethite content of 10% (line 251), but the SEM/EDS and EMP results show much higher Fe values. This suggests to me that the XRD patterns are failing to show the presence of non-crystalline iron minerals. This is another topic that requires some discussion.

Response: Thanks again, this is another excellent comments and questions, I would like to address too. XRD data have too much signal of quartz. We try to remove the quartz background to get better XRD data for limonite family minerals. This is can be done by sample preparation for goethite-rich spherules (Lee and Xu, 2016, and Lee et al. 2016). We try many times, however, still a lot of signal of quartz were presented. We have not figure out why we can not remove quartz signal easily. We guess the reason is that the quartz is too small too. We need do more work to test this idea.

The XRD calculations of 10% is not accurate. We explain in text in line 249-250 why we get so low content in XRD data. The reason is that the quartz signals or peaks are too strong, and goethite are too small and the signal were not so good. SEM, EMP and TEM give more accurate value of percentage.

  1. Title:  should refer to Tiahu Lake, China so that international readers will clearly know the location.

Response: Agree, we like the suggestion. We change title in line 3. Add “in China” after “Taihu Lake Basin”. 

  1. Lines 28, 52: the reference to siderite concretions is confusing, since the masses described in this report contain very little siderite. I suggest a better description would be “ferruginous concretions”.

Response: Agree, “siderite” was changed to “ferruginous” in line 51. Siderite concretion is another story, we have another paper to deal with siderite concretion which also consist with our hypotheses of airburst plume. It is right we do not need to confuse the readers, otherwise we need more words to explain.

  1. Line 67: “most abundant” should be “the most abundant”.

Line 66: “most abundant” was changed to “the most abundant”.

  1. Line 113: “Drilling” should be “drilling”.

Line 132: “Drilling” was corrected as “drilling”.

  1. Line 114: “agate” is a mineral name. I suspect the correct term here is “agate mortar” or “agate grinding chamber”.

Line 140, 141: The statement “agate” was corrected as “agate grinding chamber”.

  1. Line 166: “diffraction patter” is a typing error for “diffraction pattern”.

Line 165: “diffraction patter” was corrected as “diffraction pattern”.

  1. Line 174:  The size range is needed for the irregular or massive concretions, to match the description of other concretion forms.

Response: Agree with that suggestion that the size range of the irregular or massive concretions is needed. “(size from 1cm to 15cm)” was added after “massive concretions” in Line 173, 174.

  1. Lines 138, 190, 233: “Bean sized” is an uncertain description, because various bean species have a very wide size range.  In line 448, the description is “Bean-mm-sized”. I suggest eliminating “bean” as a size category, and instead make reference to actual metric dimensions.

Response: Agree the Reviewer’s suggestion, the statement “Bean-sized” was change to “mm-sized” in Line 137, 189, 217, 231, 242, 445.

  1. Line 405 refers to “ginger-shaped” concretions, a description that will lack meaning for many readers.

Response: “ginger-shaped” concretions looks like ginger. Many calcareous nodules called ginger stone were found in loess layer. We change the ginger-shaped to “ginger stone” which is common geology term in line 402.

  1. Line 254: the wording needs to be re-considered.  As presently phrased, the statement is that the signals from quartz are too strong to reduce the signals from goethite. This is the reverse of the intended meaning: the strong quartz signals reduce the signals detected from goethite.

Response: Agree and made changes in Line 252, 253.

  1. Table1:  The values need to be labelled as “weight %”, not “%”

Line 326: “weight” was added.

  1. Line 377: “TOC” is described as an abbreviation for “total of carbon”. The correct definition is “total organic carbon”.

Response: We are sorry for this mistake. The “total of carbon (TOC)” in line 357 is wrong. It should be “Total Organic Carbon (TOC) in line 357. We corrected this.   “total organic carbon” is abbreviated as “TOC” which are in many literatures. Line 376’s abbreviation is correct, do not need change.

  1. Line 451: “kind of topological surfaces…..” would be clearer with something like “The surface topography of spherules varies from smooth curved surfaces (Fig. 4) to uneven surfaces with many pits (Fig. 4c).”

Response: Thanks for suggestion. As Reviewer suggested that we have re-written this part. “The spherules have kind of topological surface with curved smooth surfaces (Fig 4). Some have uneven surfaces with many pits (Fig. 4c).” was changed to “The surface topography of spherules varies from smooth curved surfaces (Fig. 4) to uneven surfaces with many pits (Fig. 4c).” in Line 447, 448, 449.

  1. Line 471: “more loose” is a questionable descriptions. Perhaps you mean “more porous”?

Line 467: Thanks. We did change as reviewer suggested that “more loose” was corrected as “more porous”.

  1. Line 486: “certain amount spherules” should be “certain amount of spherules”.

Line 483: “of ” was added.

  1. Line 494: “we hypotheses” should be “we hypothesize”.

Line 491: “hypotheses” was corrected as “hypothesize”.

  1. Figure 4: The yellow scale bars for images a,b,c are difficult to read. Changing to black color would help, in addition to increasing the font size.

Response: Thanks for this very helpful suggestion. We made changes of scales in Figure 4.a, b, c, and increase the font size to readable for readers. See page 7 for details.  

Reviewer 2 Report

Dear Authors:

The present manuscript called "Iron-Rich Spherules of Taihu Lake: Origin Hypothesis of Taihu Lake Basin" presents novel information and is very easy to understand. I have carefully reviewed the document, and it is well written, structured, and the methodology, results and discussions are very clear.

I recommend the manuscript for publication

Some slight details to improve:

Line 31. Correct 65-km… 65 km
Line 62. Correct (Zuo and Xie, 2020)… Metals format
Line 135. 200 g
Line 137- 50 μm
Review the separation between the number and the international unit used throughout the document.

Regards

Author Response

Open review #2:

  1. Line 31. Correct 65-km… 65 km

Line 31: “65-km” was corrected as “65 km”.

  1. Line 62. Correct (Zuo and Xie, 2020)… Metals format

Line 61: Metals format was corrected. “(Zuo and Xie, 2020)” was changed to “[14]”.

  1. Line 135. 200 g

Line 134: “200g” was corrected as “200 g”.

  1. Line 137- 50 μm
    Line 137: “50-μm” was corrected as “50 μm”.
  2. Review the separation between the number and the international unit used throughout the document.

Response: We are very sorry for our negligence of the separation between the number and the international unit. We have reviewed the document and corrected them. See the paper for details.

Reviewer 3 Report

  1. perhaps you need to pay attention to the design. There are few errors (like lines 373 and 374).
  2. Tell me why you did not find it important to give the results of the XDR analysis of the silty layer, besides, you have it. (as you indicate in the article).
  3. spherical nodules enriched in Mn are usually confined to the apical parts of zones of stable Holocene nepheloid accumulation. They usually occur in a thin layer of reduced sandy-argillaceous sediments overlapped from the surface by a layer of argillaceous fluff - a brown oxidized film and black flowing silts with a negative (or close to it) redox potential, i.e. the spherules themselves are confined to the redox geochemical barrier. The question is whether it would be useful for you to clarify this question - with the redox potential. (The change in the redox potential in the vertical section of the cogncretion deposits directly affects the migration of manganese in the bottom and silt waters.)
  4. Also, a change in the lower boundary of the barrier Eh, may be associated with the secondary dissolution of spherules, which are observed in the work. Perhaps this will be one of the factors in the formation of spherules. Has this question been raised? 

Author Response

Open Review #3:

  1. perhaps you need to pay attention to the design. There are few errors (like lines 373 and 374).

Response:Thanks for suggestion. We did little changes to make the statement clear. Please see Line 372 to 374 for details.

  1. Tell me why you did not find it important to give the results of the XDR analysis of the silty layer, besides, you have it. (as you indicate in the article).

Response:The question I think is related to line 255-256 which said we have XRD data from silty layer which is similar to spherule sample B. The question is why we did not give the results of XRD data of silty layer.

I think the question and suggestion are very good. It is very important to know the background of these iron-rich spherules. These data of silty layer is very important to explain how iron-rich spherules formed and how these two linked. XRD data from silty layer are very similar to the profile of sample B (Fig. 5). We keep the text unchanged. 

  1. spherical nodules enriched in Mn are usually confined to the apical parts of zones of stable Holocene nepheloid accumulation. They usually occur in a thin layer of reduced sandy-argillaceous sediments overlapped from the surface by a layer of argillaceous fluff - a brown oxidized film and black flowing silts with a negative (or close to it) redox potential, i.e. the spherules themselves are confined to the redox geochemical barrier. The question is whether it would be useful for you to clarify this question - with the redox potential. (The change in the redox potential in the vertical section of the cogncretion deposits directly affects the migration of manganese in the bottom and silt waters.)

Response:Sorry, I did not understand the question and the question is not clear. I guess the question is how to explain the Mn migration under redox condition which may relate to page 14. In page 14, our discussion is to say that these spherules are obviously different from typical iron-manganese nodules. We did not address the Mn issue. Mn is linked to Fe. This is might be good point to think in future work.

Reviewer also mention one important issue, redox potential issue, which is relate to the valence state of iron, Fe3+ or Fe2+.We pay attention to this issue, however we have not figure out yet. Therefore, we put this issue aside right now, and may address this issue in future.

  1. Also, a change in the lower boundary of the barrier Eh, may be associated with the secondary dissolution of spherules, which are observed in the work. Perhaps this will be one of the factors in the formation of spherules. Has this question been raised? 

Response:Thanks for comments. Reviewer also mention one important issue, redox potential and barrier En issue, which are relate to the valence state of iron, Fe3+ or Fe2+..These conditions are very important for the formation of spherules. We think the spherules were formed in reduced environment which may related to impact plume or deep water bottom which is also in reduced condition but with more quiet environment. We pay attention to this issue, however we have not figure out yet. Therefore, we put this issue aside right now, and may address this issue in future.

Thanks again for your comments.